chemical engineering/materials science/ mathematical modelling

high-pressure separation, composite mixed matrix membranes, CO₂-CH₄ separation, Maxwell model, performance prediction

**Author for correspondence:**
Hilmi Mukhtar
e-mail: hilmi_mukhtar@utp.edu.my

This article has been edited by the Royal Society of Chemistry, including the commissioning, peer review process and editorial aspects up to the point of acceptance.

# Composite amine mixed matrix membranes for high-pressure CO₂-CH₄ separation: synthesis, characterization and performance evaluation

Nur Aqilah Bt Fauzan[1], Hilmi Mukhtar[1], Rizwan Nasir[3], Dzeti Farhah Bt Mohshim[2], Naviinthiran Arasu[1], Zakaria Man[1] and Hafiz Abdul Mannan[4]

[1]Department of Chemical Engineering, and [2]Department of Petroleum Engineering, Universiti Teknologi PETRONAS, Seri Iskandar, 32610 Perak, Malaysia
[3]Department of Chemical Engineering, University of Jeddah, Jeddah 23890, Saudi Arabia
[4]Institute of Energy and Environmental Engineering, University of the Punjab, 54590 Lahore, Pakistan

(iD) HM, 0000-0002-1862-0924

The key challenge in the synthesis of composite mixed matrix membrane (MMMs) is the incompatible membrane fabrication using porous support in the dry–wet phase inversion technique. The key objective of this research is to synthesize thin composite ternary (amine) mixed matrix membranes on microporous support by incorporating 10 wt% of carbon molecular sieve (CMS) and 5–15 wt% of diethanolamine (DEA) in polyethersulfone (PES) dope solution for the separation of carbon dioxide (CO₂) from methane (CH₄) at high-pressure applications. The developed membranes were evaluated for their morphological structure, thermal and mechanical stabilities, functional groups, as well as for CO₂-CH₄ separation performance at high pressure (10–30 bar). The results showed that the developed membranes have asymmetric structure, and they are mechanically strong at 30 bar. This new class of PES/CMS/ DEA composite MMMs exhibited improved gas permeance compared to pure PES composite polymeric membrane. CO₂-CH₄ perm-selectivity enhanced from 8.15 to 16.04 at 15 wt% of DEA at 30 bar pressure. The performance of amine composite MMMs is theoretically predicted using a modified Maxwell model. The predictions were in good agreement with experimental data after applying the optimized values with AARE % = ~less than 2% and $R^2 = 0.99$.

# 1. Introduction

Generally, the high-grade gas separation membranes for industrial applications should possess superior separation performance with outstanding permeability, which has the potential to overcome the trade-off relationship of the Robeson upper bound for polymeric membrane [1]. Moreover, membranes that are mechanically stable, cost-effective in production, and able to withstand high pressure and temperature are desirable [2,3]. To achieve these criteria, the membrane materials and structures are the critical decision as they have a profound influence on the entire membrane performance [4]. Thus, it is very important to develop a new combination of membrane materials that meet industrial requirements.

Attempts are being made by researchers to incorporate inorganic fillers such as zeolite, metal-organic framework (MOF), carbon molecular sieve (CMS), and other fillers into a polymer matrix to develop mixed matrix membranes (MMMs). The MMMs are reported to have higher gas permeabilities with recuperated gas selectivity contrary to respective pure polymer membranes [5,6]. Recently, Cheng *et al.* showed a substantially improved $CO_2$-$CH_4$ selectivity and permeability by adding coated size-selective MOF cores with a covalent organic framework (COF) in polysulfone (PSF) with a thickness of 40–70 µm [7]. They reported 48% and 79% enhancements in $CO_2$ permeability and $CO_2$-$CH_4$ selectivity, respectively, in comparison with that of neat PSF from 11.3 Barrer to 22.9 Barrer. Nonetheless, the $CO_2$ permeability of 22.9 Barrer is still below the standard for cost-effective $CO_2$ capture and Robeson upper bound curve. When the thickness of MMM is above 10 µm [8,9], this will consequently cause low $CO_2$ permeability of 0.1–22.9 Barrer and high cost of membrane fabrication. Thus, MMMs need to be made from a thinner selective layer to achieve the industrial demand for elevated product yield, which means high gas flux.

Nevertheless, the majority of MMMs are dense membranes with thickness exceeding 10 µm as a result of complication in preparing fillers of this proportion and in averting particle agglomeration during membrane fabrication [10–12]. The thin-film composite membrane (TFCM) is being introduced to overcome the challenges mentioned above. TFCMs comprise a minimum of two layers, i.e. the support layer and the selective layer [13,14]. Additionally, there may be other layers such as the gutter layer to diminish pore penetration and protective layer to overcome pore defects [15]. Norahim *et al.* developed TFCM by incorporating graphene oxide (GO) in polyethylene glycol (PEG 400)/polyether block amide (Pebax 1657) blended polymer to form a selective layer on polyetherimide (PEI) with a non-woven backing support layer (Novatexx 2470) [16]. The 50 : 50 ratio of PEG/Pebax membrane, including the non-woven support layer, resulted in a double increment of permeance and $CO_2$-$CH_4$ selectivity compared to pure Pebax 1657. In another work, Khalilinejad *et al.* also prepared three-layer TFCM of Pebax 1657, polyvinyl chloride (PVC) with non-woven support layer, and reported that the $CO_2$ permeance and $CO_2$-$CH_4$ selectivity of the membrane increased upon pressure increment with 16.7% and 21% in contrast to the pure Pebax 1657 [17]. Furthermore, Mozafari *et al.* reported the use of a non-woven support layer with polymethylpentyne (PMP) and 2 wt% UiO-66-$NH_2$ and formed a defect-free boundary which managed to increase $CO_2$-$CH_4$ selectivity from 15.9 to 35.3 at 5 bar [18].

Several modelling approaches have been developed to minimize the time-consuming and costly experiments and to predict the performance of MMMs and MMMs with a third component [19–23]. On the other hand, the performance of membranes depends upon the operating parameters like pressure and temperature. As the pressure difference between feed and permeate side is large, the membranes exhibit high permeability [24]. Therefore, it is necessary to model the effect of pressure on the performance of MMMs. In the literature, very limited studies are reported to address the pressure dependency of gas transport. Maghami *et al.* [21] modified the Van't Hoff-Arrhenius equation to model the transport of gas through MMM, using the corresponding experimental data at 35°C in the feed pressure range from 2 to 12 atm. They found 5.1% AARE for 300 data points.

In our previous work, we have reported high $CO_2$ permeance and $CO_2$-$CH_4$ selectivity up to 117.32 GPU and 20.21, respectively, polyethersulfone (PES) with different concentrations of DEA and fixed loading of carbon molecular sieve (CMS) [25]. However, these membranes were not tested at high pressure due to the possibility of fracture of the membrane in high pressure without any porous support layer. Building on this earlier work, in this study, we have added a non-woven support layer at the bottom layer of the amine MMM. To the author's best knowledge, no study on the synthesis and performance analysis of composite amine MMM at high pressure is available. Moreover, there is no study reported in the literature on the performance prediction of composite amine mixed matrix membranes. Therefore, the amine composite mixed matrix membrane is developed and characterized

**Table 1.** Membrane samples composition.

| polymer concentration (wt%) polyethersulfone (PES) | CMS loading (wt %) | DEA amine concentration (wt%) | membrane ID |
|---|---|---|---|
| 20 | 0 | 0 | CM |
| 20 | 10 | 0 | CM-C10 |
| 20 | 10 | 5 | CM-C10D5 |
| 20 | 10 | 10 | CM-C10D10 |
| 20 | 10 | 15 | CM-C10D15 |

CM, composite membrane; C, carbon molecular sieve; and D, di-ethanolamine.

for high-pressure application up to 30 bar. The permeance and ideal selectivity of the synthesized MMMs were investigated. Also, the Maxwell Model was used because it provides the exact solution for the permeability of randomly distributed and non-interactive homogeneous solid spheres in a continuous matrix [26,27]. The model is modified by considering the effect of high pressure (10–30 bar), filler loading (10 wt%), and amine concentration (15 wt%) on the carbon dioxide transport properties of PES-CMS-DEA composite amine MMMs.

# 2. Experimental procedures

## 2.1. Materials

The flaked form of PES polymer (ULTRASONE E-6020P) was purchased from BASF Germany. The average molecular weight of PES was 50 000 g/mol. PES was the main polymer for polymeric membranes fabrication in this study because of its advantageous properties, including chemical, thermal and mechanical stability [28]. The glass-transition temperature ($T_g$) for PES was 225°C. The solvent $N$-methyl-2-pyrrolidone (NMP) used for this work was purchased from Merck, Germany, with informed purity of 99.99%. NMP was used as a solvent because of its high chemical stability, solubility, low flammability, boiling point, and low volatility, which is good for the homogeneous membrane's outcome. The CMS is used as an inorganic filler in the composite MMMs synthesis, and it allowed fast transport of gas molecules and had a uniform pore size on the surface [29]. It was acquired from Enviro Chemicals, Japan. DEA with 99% purity was purchased from Merck. DEA is the most extensively used secondary amine for the removal of acid gases, due to its favourable reaction kinetics and resistance to solvent degradation [30]. The support layer used at the bottom of the composite amine mixed matrix membrane was a Novatexx 2471, acquired from Freudenberg Filtration Technologies, Germany.

## 2.2. Methods

Five different types of membranes were synthesized in this study. The list of membranes synthesized, along with their concentrations, is tabulated in table 1. The polymer concentrations reported in this study were selected based on dope solution viscosity which has been studied in our previous study [31].

### 2.2.1. Synthesis polyethersulfone composite membrane

The procedure for the synthesis of pure PES composite membrane consists of (i) dispersion of the PES in NMP, (ii) casting of the solution onto a non-woven fabric, and (iii) successive dry–wet phase inversion in the non-solvent water bath. The based PES composite membrane was synthesized by using 20 wt% dried PES of solvent basis at critical polymer concentration. The dried PES was added into the NMP solvent and stirred (100 rpm) for 24 h at room temperature. The entrapped air bubbles in the solution were removed by degassing in the Sonication bath (Transsonic Digital S, Elma) for 1 h. Membranes were cast using polypropylene, non-woven fabric as a support layer on a flat, dust-free, dry, smooth, glass

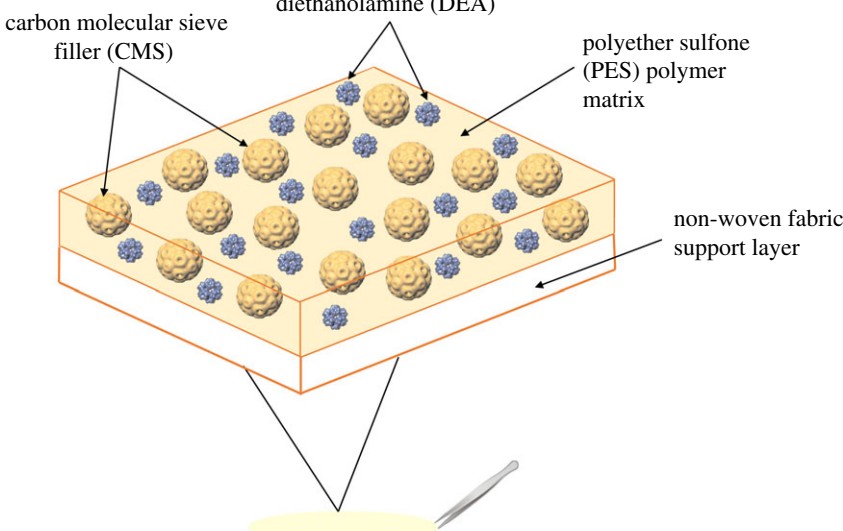

diethanolamine (DEA)

carbon molecular sieve filler (CMS)

polyether sulfone (PES) polymer matrix

non-woven fabric support layer

mixed matrix membrane + thin composite layer

**Figure 1.** Representation of amine composite mixed matrix membranes.

plate. Water and acetone were used to wash the glass plate and remove the impurities. Dust particles were removed from the glass plate by using compressed air. The mixed polymer solution was transferred onto the glass plate with an adjusted casting knife thickness of 200 µm. Then, the casting knife was relocated over the solution to form the membrane's uniform layer by using a film applicator. The prepared membrane film was dried for about 60 s at room temperature by dry phase inversion before it was immersed in water for wet phase inversion. The wet phase inversion technique used water as a non-solvent [32] to convince a chain of liquid–liquid phase separations [33]. As the coagulation step completed, the TFCM was washed using distilled water and placed inside the room temperature condition to left dry. The synthesized membranes were to be used for characterization.

### 2.2.2. Synthesis of composite mixed matrix membrane

The procedures of casting composite mixed matrix membrane consisted of (i) dispersion of the CMS particles in NMP solvent, (ii) sonication of the solution to keep particles in suspension, (iii) mixing of PES with CMS solution, (iv) casting of the solution onto a non-woven fabric and (v) successive dry–wet phase inversion in the non-solvent water bath. The 10 wt% CMS particles were added in NMP and stirred for 15 min. The CMS-NMP solution was ultrasonicated for 1 h at a frequency of 100 Hz. 20 wt% of dried PES on a solvent basis was added to the solution and stirred for 24 h at room temperature. The entrapped air bubbles in the solution were degassed for 1 h. After degassing, the mixed matrix membrane solution was cast as described in §2.2.1. The prepared membrane film was dried for about 60 s by dry phase inversion before it was immersed in water for wet phase inversion. As the coagulation step completed, the TFCM was washed using distilled water and placed inside the room temperature condition to dry.

### 2.2.3. Synthesizing of composite amine mixed matrix membrane

Three composite amine mixed matrix membranes were synthesized by using the same methodology, as discussed in §2.2.2. The procedures of casting composite amine mixed matrix membrane consist of (i) dispersion of the CMS particles in NMP solvent, (ii) sonication of the solution to keep particles in suspension, (iii) mixing of PES and DEA with CMS solution, (iv) casting of the solution onto a non-woven fabric and (v) successive dry-wet phase inversion in the non-solvent water bath. The 5 wt%, 10 wt% and 15 wt% amine compositions based on the polymer weight were used in this study. The schematic illustration of amine composite mixed matrix membranes is shown in figure 1. The process for membrane development is visualized in figure 2.

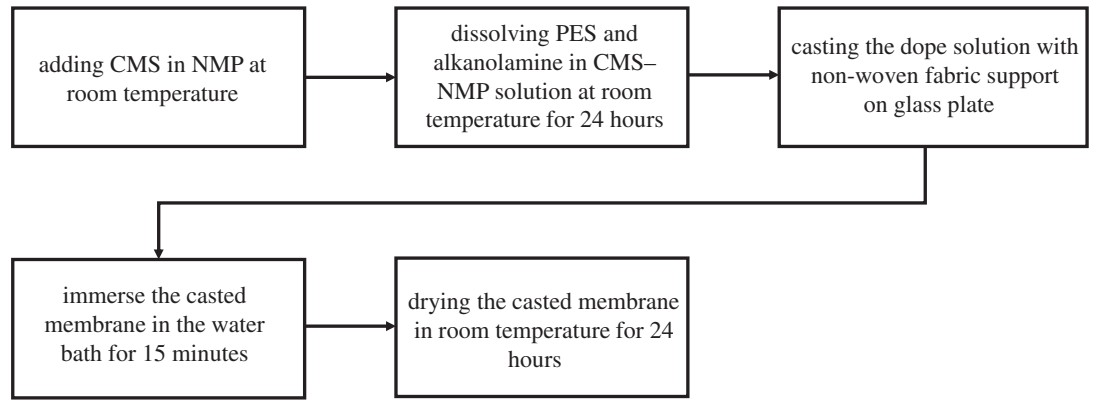

**Figure 2.** Process for the membrane's preparation methodology.

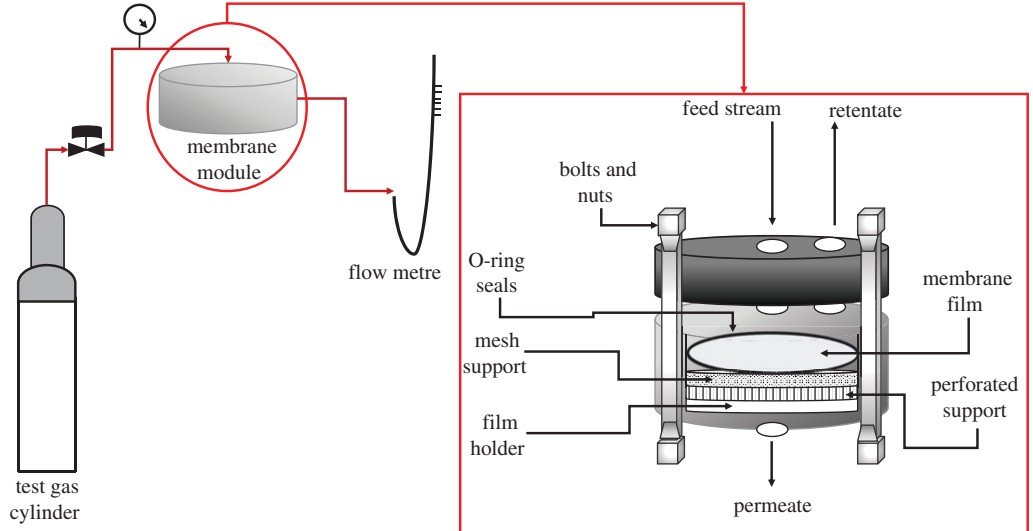

**Figure 3.** Gas permeation test set up.

## 2.3. Characterization of membranes

The synthesized membrane had been analysed for physico-chemical characterization. Listed below are the physico-chemical properties of membranes to be characterized and analysed. The variable pressure field emission scanning electron microscopy (VPFESEM) is used to study the morphology of the composite membrane with polypropylene non-woven fabric evaluated. The Fourier transform infrared (FTIR) is used to examine the functional group in the composite membrane between polymer components and inorganic materials. The differential scanning calorimetry (DSC) is used to find the glass transition temperature ($T_g$) of membranes. It provides a qualitative analysis of the flexibility or rigidity of the polymer chain. The effect of polypropylene non-woven fabric on mechanical properties of amine mixed matrix membrane was evaluated by universal testing machine (UTM).

## 2.4. Gas separation analysis

The membranes were tested for the permeance of $CO_2$ and $CH_4$ (99.99% purity), using a gas permeation unit at a feed pressure of 10, 15, 20, 25 and 30 bar at room temperature (25°C). The gas permeation unit is described as a 17.35 cm$^2$ two-compartment cell. The membrane was arranged on a porous support, holding O-rings between two flanges. The unit was vacuumed to eliminate any residual gases. Permeate pressure was constant at atmospheric pressure, P atm. The volumetric flow rate of permeate gas streams was calibrated using a bubble flow metre. Figure 3 represents the schematic set up of the gas permeation equipment [25,34].

cross-sectional view                                                    top view

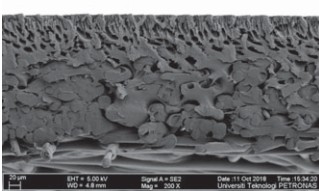
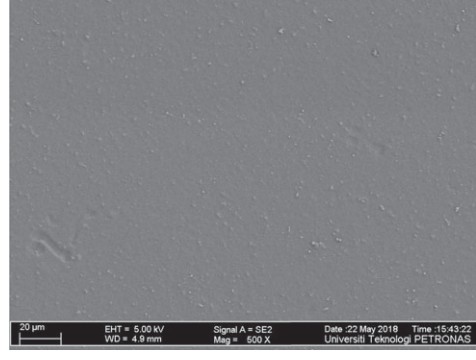
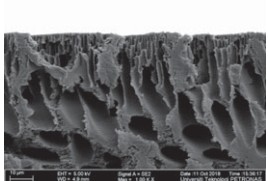
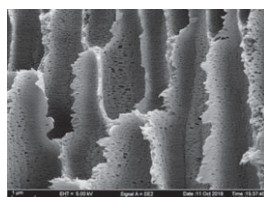

**Figure 4.** Cross-sectional and top view of pure PES composite membrane.

A bubble flow metre which can detect flow rates as low as $100\,\mathrm{ml\,mi^{-1}}$ was used to measure the permeate rate of gas streams. The equation used to calculate the gas permeance in GPU (1 GPU = $1 \times 10^{-6}\,\mathrm{cm^3\,(STP)/s.cm^2.cmHg}$) is as follows:

$$\frac{P_i}{L} = \frac{J_i}{\Delta p_i} \tag{2.1}$$

where $L$ is the membrane thickness (cm), $J_i$ represents the gas flux ($\mathrm{cm^3\,cm^{-2}\,s}$), whereas $\Delta p_i$ is the transmembrane pressure (cmHg). Instead, the selectivity of the membrane was calculated by the following equation:

$$\alpha_{\mathrm{CO_2/CH_4}} = \frac{P_{\mathrm{CO_2}}}{P_{\mathrm{CH_4}}}. \tag{2.2}$$

# 3. Results and discussion

## 3.1. Morphological analysis of synthesized composite mixed matrix membranes

The morphological analysis is carried out for all synthesized membranes, and for the ease of understanding discussion about morphologies of pure PES, composite mixed matrix membranes and amine composite mixed matrix membranes are divided into subsections. As described in the above section, the synthesis of all composite membranes has been carried out by phase immersion precipitation by using water as a non-solvent. It is, therefore, necessary to understand the effect of the fabrication technique on the membrane structure because the membrane structures are a major contributor to the efficiency of membrane separation [35,36]. The membrane structure may be categorized based on polymer, solvent, and non-solvent combination. In the cited literature, it is mentioning that the membranes developed by a combination of solvent (NMP)/(non-solvent water) have a finger-like and spongy structure [37,38]. All developed composite membranes exhibit the finger-like and spongy structure. Therefore, for ease of understanding, the explanation of the structures has been discussed here.

### 3.1.1. Morphological analysis of pure polyether-sulfone composite membranes

The top and the cross-sectional view of the PES composite membrane was analysed by FESEM, as visualized in figure 4. It is observed that the surface of the developed membrane is smooth, dense and defect-free. The cross-sectional view of the PES composite membrane has shown the asymmetric structure, which was composed of non-woven fabric with a finger-like and spongy structure. The developed pores in finger-like structure are sponge pores. Moreover, the top view of the pure PES membranes is smooth and non-porous. The structure was produced because of the immediate phase inversion between solvent and non-solvent in the coagulation bath. Besides, the strong affinity

cross-sectional view

top view

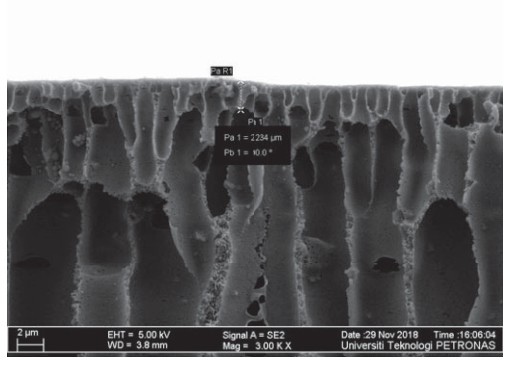

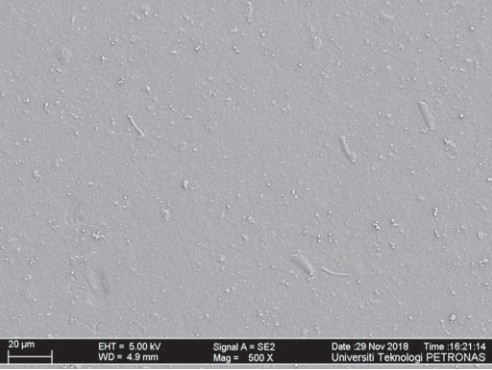

cross-sectional view

cross-sectional view

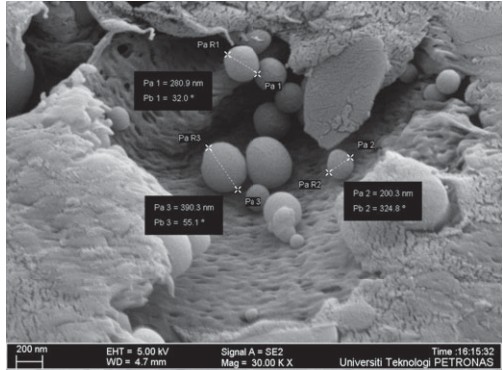

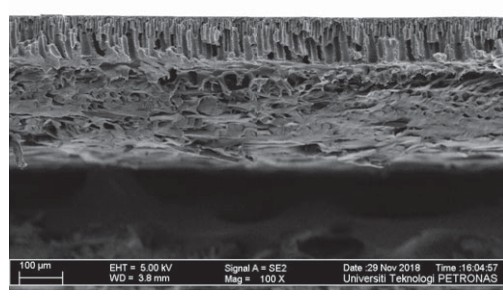

**Figure 5.** Top and cross-sectional view of composite mixed matrix membrane CM-C10.

between NMP and water caused the formation of such a structure [39,40], and this type of structure affects the mechanical strength compared to a membrane structure containing macro voids [41].

### 3.1.2. Morphological analysis of composite mixed matrix membranes

The top and cross-sectional micrographs of CMS (10 wt%) loaded polyethersulfone composite membrane (CM-C10) are shown in figure 5. The top view of the developed membrane is smooth and without any pinholes and micro-voids. The cross-sections of supported CM-C10 composite membranes exhibit the 183.1 μm thickness of non-woven fabric support and 93.86 μm thickness of CM-C10 above the support. Good compatibility between CM-C10 and non-woven fabric support was also observed. Moreover, in the cross-section micrographs, the CMS particles are well distributed with very little agglomeration.

The smooth surface of CM-C10 is due to the carbon-based structure of CMS, the dispersion of CMS is good enough, and no agglomeration was observed on the surface of the CM-C10 mixed matrix membrane. Moreover, the absence of cracks on the surface shows that the developed CM-C10 is not brittle and has good stability [42]. The cross-sections of CM-C10 exhibit the typical asymmetric porous structure with no microvoids. The dispersion of the CMS caused competition between the shearing forces that were applied to break the agglomeration and the coherent force that keeps the agglomerate from dispersing. Therefore, the cohesive force that arises from the Vander Walls interparticle interaction between CMS molecular sieves becomes dominant at higher loading than the shearing forces that attempted dispersion of CMS particles [43]. These results are supported by previous studies where the structure of fabricated membrane samples showed a high similarity [44,45].

### 3.1.3. Morphological analysis of composite amine mixed matrix membranes

The top and cross-sectional morphology of ternary composite amine mixed matrix membranes with 5 wt %, 10 wt% and 15 wt% DEA and 10 wt% CMS is given in figure 6a–c. All top views are smooth and

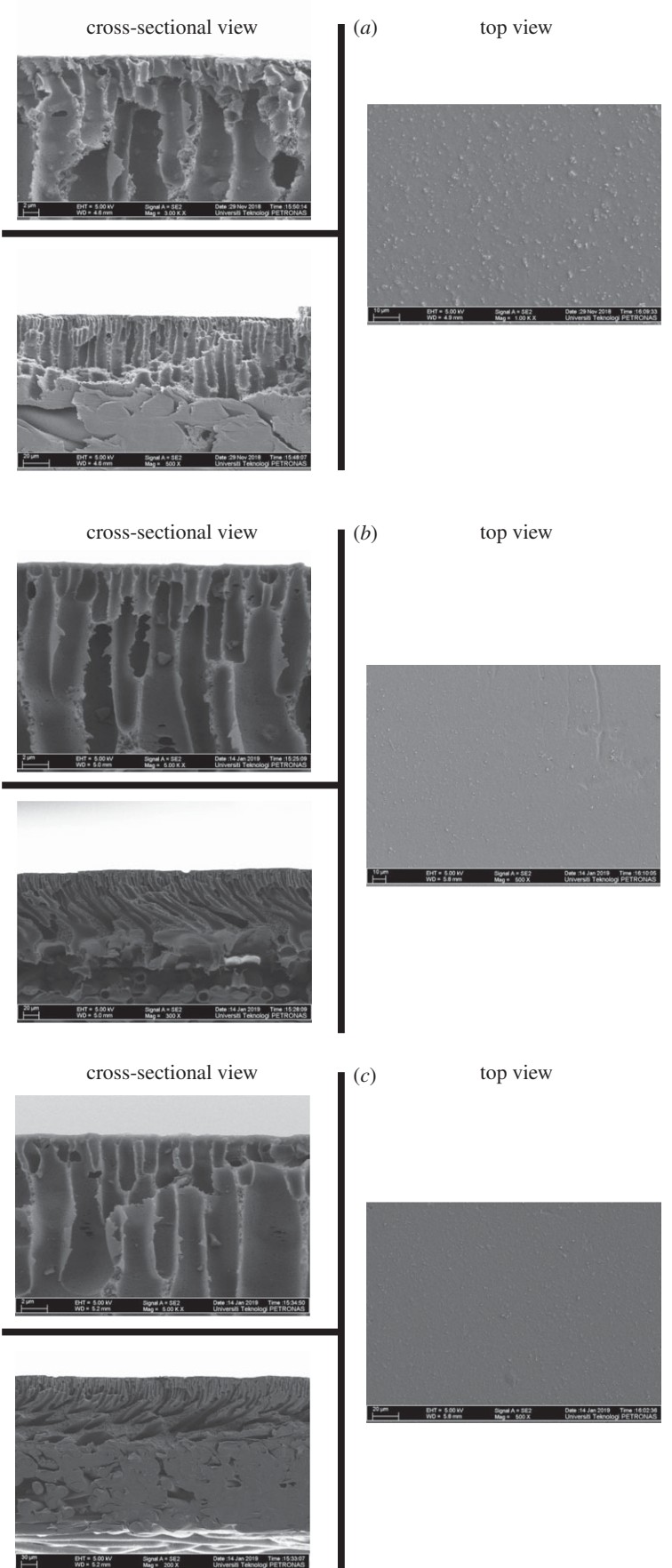

**Figure 6.** Top and cross-sectional view of amine composite MMMs with 10 wt% CMS and (*a*) 5 wt% DEA, (*b*) 10 wt% DEA and (*c*) 15 wt% DEA.

**Table 2.** Possible FTIR spectra bonds in developed composite membranes [49].

| possible spectra bonds | band frequency (cm$^{-1}$) |
| --- | --- |
| C=C aromatic | 1575.18 |
| C=O ketone | 1730.08 |
| benzene/ aromatic rings | 1484.00 |
| sulphone oxide | 1020.55 |
| C-O-C ether | 1139.65 |
| —CH$_2$ bending | 2928.55 |
| —O-H | 3349.99 |
| —N-CH$_3$ | 2928.55 |

defect-free and cross-sectional micrographs exhibit the typical finger-like and sponge-like pore structures, which help to reduce the transport resistance and to provide enough mechanical strength, respectively. The cross-sections of CM-C10D5, CM-C10D10 and CM-C10D15 exhibit the 203.0 μm thickness of non-woven fabric support and 34.99 μm, 126.90 μm and 137.50 μm thickness of CM-C10D5, CM-C10D10 and CM-C10D15, respectively, above the support. It was also observed at higher magnifications that by the addition of low molecular DEA, the small agglomeration which appeared in CM-C10 membranes had disappeared, and CMS is well dispersed in the PES matrix. The good dispersion of CMS by the addition of DEA is because of the good interaction and solubility of PES, NMP and DEA. DEA helped CMS particles to be wetted and surrounded better with PES chains. It is observed that with the addition of DEA, the CMS particles have not shown apparent defects or agglomeration and void formation. It was feasible to make interfacial void-free CMS-filled glassy polymer membranes by filling the CMS-polymer interface with a low molecular weight third component, which could interact simultaneously with polymer and CMS. The strongest interaction can be hydrogen bonding between polymer and the third component [46]. The absence of phase separation in the PES-CMS-DEA membrane is due to the solvency of the third component in the solvent during the dope solution preparation, as suggested by Yong *et al.* [46]. As the DEA also belongs to a low molecular weight additive family, the addition of DEA makes the dope solution less stable (thermodynamically), which causes rapid demixing in the coagulation bath. The nature of the additive may affect the exchange rate of solvent and non-solvent during phase inversion process and influence the kinetic of precipitation and formation of resulting membrane morphology [47]. The change in kinetics has a negligible effect because a very low amount of additive has been added to cause a change in the viscosity of the dope solution [48]. Thus, it is expected that the presence of DEA, improved membrane structure and non-woven support will enhance the performance of the synthesized membranes at high pressure.

## 3.2. Spectral analysis

The analysis is carried out to assess the possible change in polymer (PES) structure or any chemical interaction due to the addition of DEA and inorganic filler CMS. The FTIR analysis of all developed composite membranes is recorded and shown in electronic supplementary material, figure S1. It was observed that all spectra are almost identical to each other and concluded that there is no chemical interaction between PES and DEA. It is also observed that by the addition of CMS, there is no significant change in the PES spectra, which confirms the physical attachment of the polymer chain to CMS. But by the addition of DEA, there were small increases or decreases in the band frequencies, as shown in electronic supplementary material, figure S1. The explanations of the functional group of the possible peaks are tabulated in table 2. The findings of this study are similar to the previously cited literature [25,50,51].

## 3.3. Thermal analysis

### 3.3.1. Differential scanning calorimetry analysis

DSC analysis was carried out to study the effect of CMS and DEA addition on the glass transition temperature of synthesized membranes. The glass transition temperature of neat CM, CM-C10 and

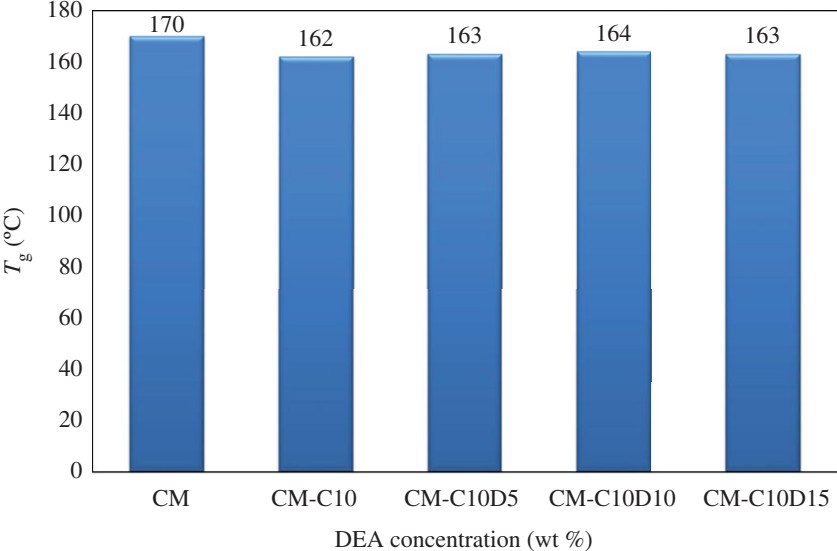

**Figure 7.** Glass transition temperatures of developed composite membranes.

CM-C10D (5–15) with fixed loading of CMS (10 wt%) and various concentrations of DEA (5–15 wt%) composite membranes was tabulated in table 2. It was observed that at lower loading of CMS (10 wt%), the $T_g$ is decreased. This may be due to an increase in free volume and chain mobility around CMS. Thus, the $T_g$ of the PES shifts to lower temperatures with the addition of CMS particles [52,53].

Figure 7 shows the effect of DEA on $T_g$. It was noted that the decrease of about 6–8°C below that of pure PES composite membrane (CM) at various concentrations of DEA (5–15 wt%) was observed. The $T_g$ of CM-C10D (5–15 wt%) amine composite MMM is lower than the $T_g$ of pure PES composite membranes. It is also observed by the addition of DEA in membranes, and the $T_g$ is reduced. This shows that membranes are less crystalline and have an amorphous structure. The addition of DEA decreases the value of $T_g$ and increases the permeance of the membranes. It is believed that DEA is a low molecular weight organic solvent added to a polymer matrix to modify its physical properties such as flexibility (by lowering the glass transition temperature) and microstructure. The improvement of the structure has been confirmed through morphological analysis. Furthermore, DEA is relatively small as compared to that of the polymer molecule, and henceforth can easily penetrate the polymer matrix and enhance the interaction of membrane phases, and may reduce the cohesive forces operating between the polymer chains increasing the chain segmental mobility [54]. The variation of $T_g$ has been correlated with the gas permeability for different concentrations of additive, which will be discussed in §3.5.2. Therefore, it can be concluded that the addition of a DEA to the formulation of a mixed matrix membrane strongly affects its final structure. Sen *et al.* also found a similar trend for polycarbonate-zeolite 4A and pNA membranes. They found that the addition of pNA in PC-zeolite membranes resulted in a reduction in $T_g$ values compared to pure PES membranes. They concluded that pNA provided the interaction between polymer and zeolite. Thus, according to the above study, DEA acted as an arbitrating agent to provide the interaction and is essential for CMS to affect the PES polymer matrix [55]. For all types of membranes, whether PES-CMS or PES-CMS-DEA membranes, a single $T_g$ was observed, which points to the presence of a single homogeneous phase in membranes.

## 3.4. Mechanical analysis

The mechanical properties of selected membranes were determined by tensile testing from the universal testing machine (UTM). Young's modulus and tensile strength are summarized in table 3. It was observed that by the addition of CMS, Young's modulus values are decreased from 1280 MPa to 412 MPa, which is most likely due to the stress concentration tempted by CMS aggregation at 10 wt% CMS loading. Similar findings have been observed by Manawi *et al.* [56]. On the other hand, with the addition of 5 wt% a much lower reduction (approx. 11.36%) in Young's modulus was observed, which may be due to any agglomeration of CMS. It was also

**Table 3.** Mechanical properties of synthesized composite membranes.

| membrane | Young's modulus (MPa) | tensile strength (MPa) |
|----------|----------------------|------------------------|
| CM | 1280 | 11.94 |
| CM-C10 | 412 | 12.92 |
| CM-C10D5 | 1136 | 20.08 |
| CM-C10D10 | 3278 | 17.25 |
| CM-C10D15 | — | 16.72 |

observed in table 3 that the highest elastic stiffness Young's modulus value is 3278 MPa with the addition of CMS and DEA in the polymer matrix.

Moreover, the results appeared to be independent of the loading of CMS and concentration DEA. However, Young's modulus values are independent but with the addition of 10 wt% DEA Young's modulus value is increased from 1280 MPa to 3278 MPa. The increase in Young's modulus is also due to the increase of polymer chain entanglement with CMS and DEA contents, which leads to constrained polymer chain mobility resulting in a stiffer film. It is further strengthened by the presence of polar -$SO_2$ −, DEA group and rigid aromatic ring in PES backbone. It seems to suggest that Young's modulus is likely to be underestimated from the tensile test. This claim was highlighted in previous literature [57,58].

The tensile strength is increased from 11.94 MPa to 12.92 MPa, which suggested that the addition of CMS increased the membrane strength. The increase of membrane strength may be due to restriction of chain segmental mobility by the addition of CMS [59]. It was also observed that with the addition of different concentrations of DEA, the tensile strength is enhanced as compared to pure PES membranes. This indicates that DEA is well mixed within the dope solution and effectively improves the mechanical properties of PES-CMS-DEA composite mixed matrix membranes. The improved mechanical properties confirm the strong interaction between the components of dope solution [60]. Moreover, the decrease in tensile strength with the increase of DEA concentration may be due to an increase in the flexibility of the polymer chain. The decrease in tensile strength by the increase of DEA concentration is due to the increased porosity of the developed PES/CMS/DEA composite mixed matrix membranes [47,61]. Similar findings have been observed by Ali *et al*. [62] in cellulose acetate/glycol/zinc oxide blend membranes. However, the PES/CMS/DEA composite membranes have higher tensile strength than pure PES and PES/CMS composite mixed matrix membranes. These improvements in mechanical properties of the membrane may be attributed to the strong interactions between the polyethersulfone matrix, CMS, and DEA and homogeneous dispersion and adhesion of CMS as observed by microscopy measurements.

## 3.5. Gas performance analysis

### 3.5.1. Effect of pressure on the gas separation performance

The effect of pressure (10–30 bar) on the carbon dioxide permeance and $CO_2$-$CH_4$ selectivity through composite and ternary composite mixed matrix membranes has been shown in figure 8a,b. The presented values have been calculated by equations (2.1) and (2.2). The figures exhibit a reduction in permeance and an increase in selectivity with the increase in pressure. In figure 8a, the maximum permeance is at a lower pressure of 10 bar, and as pressures approach 30 bar, the permeability is moved towards minimum value. The percentage decrease was in the range of 17.89% ± 1 to 35.90% ± 1 for all developed pressures across the entire pressure range. Figure 8b shows the opposite trend of $CO_2$-$CH_4$ selectivity. The maximum selectivity value (16.04) was at 30 bar pressure. In the cited literature, the plasticization pressure of PES is approximately 28 bar [63]. Therefore, to investigate the effect of DEA on the plasticization, the 30 bar pressure was selected. On the other hand, for all membranes, a decrease in $CO_2$ permeability with elevated pressure is observed up to 25 bar, which indicates that there is no significant plasticization [64]. After the gas had passed the plasticization pressure, it was observed that the permeance was slightly increased. This increase is due to slight plasticization [65,66]. Thus, it is concluded that the DEA has a minor effect on plasticization after the

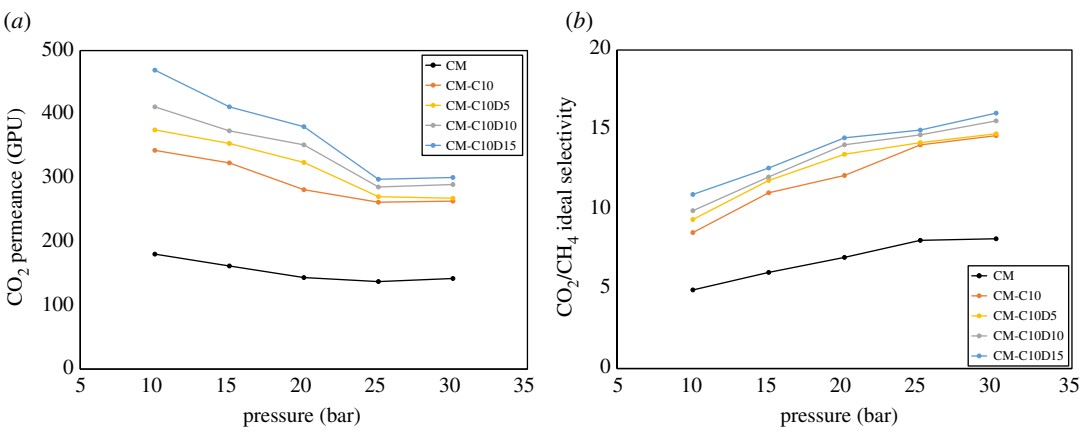

**Figure 8.** (a,b) Effect of pressure on the gas separation performance of amine composite mixed matrix membranes.

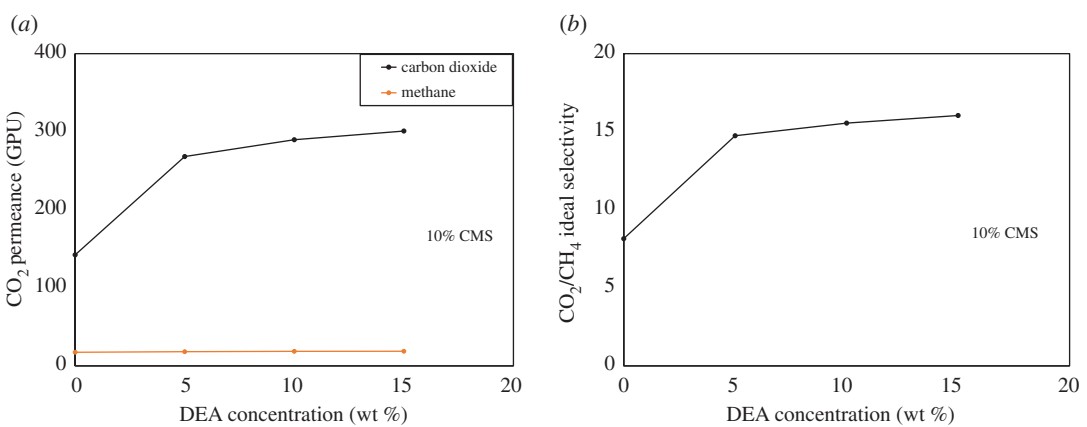

**Figure 9.** (a,b) Effect of DEA concentration on gas separation through amine composite MMMs.

plasticization pressure of the PES. The decrease in permeance before the plasticization pressure is due to competitive sorption, sorption isotherm concave shape [67,68]. It constitutes a reduction in the driving force of transport with increasing pressure and a gradual saturation of the material which may cause lower mobility [69].

### 3.5.2. Synergetic effect of carbon molecular sieve and diethanolamine addition on gas performance

The synergetic effect of carbon molecular sieve and DEA on the permeability and ideal selectivity of composite amine mixed matrix membranes at a pressure of 30 bar has been shown in figure 9a,b. The focus of this study was to investigate the effect of different concentrations of DEA with a fixed loading of CMS. According to the hypothesis, it was observed in the developed composite ternary MMMs that $CO_2$ permeance (142.21–300.84 GPU) and $CO_2$-$CH_4$ selectivity (8.15–16.04) increased by increasing the concentration of DEA as shown in figure 9a,b. By contrast, there was a very small change in the value of $CH_4$ permeance with the addition of DEA and CMS.

It was also observed that the loading of CMS was also one contributor to performance enhancement. In figure 9a, it was also observed that carbon dioxide permeance is higher than methane permeance. The pore size of CMS might be the reason for this occurrence. The kinetic diameter of $CO_2$ (3.3 Å) is smaller than the pore size of the CMS, whereas the kinetic diameter of $CH_4$ (3.8 Å) is bigger than the pore size of CMS. Therefore, it could hinder $CH_4$ molecules from passing through the membrane [70]. CMS gas transport is less resistive than the PES matrix and offers the most selective route since CMS can distinguish between the size and shape of gas penetrators. Conversely, the gaps or voids enable the bypass of gas through its non-selective and non-resistive pathways, and it is presumed to be the Knudsen diffusion [71].

The increase in the permeance and selectivity is due to 'like dissolves like phenomena', as the DEA is acting as a facilitator for the $CO_2$ transport through composite ternary MMMs. The DEA also has a good

**Table 4.** Performance comparison with literature data for $CO_2$-$CH_4$ separation.

| polymer | molecular sieve | solvent/additive | pressure (bar) | $CO_2$/$CH_4$ selectivity | reference |
|---------|-----------------|------------------|----------------|---------------------------|-----------|
| PES | MWCNT | NMP | 4 | 19.57 | [76] |
| Elvaloy4170 | functionalized multi-walled carbon nanotubes (3 wt%) | toluene | 2 | 6.18 | [77] |
| Pebax | — | glycerol triacetate (GTA) 40 wt% | 4 | approximately 13 | [78] |
| PES | CMS (10 wt%) | NMP/15 wt% DEA | 30 | 16.04 | this study |

affinity with carbon dioxide and a negligible affinity with methane gas. The escalation in $CO_2$ permeance with increasing DEA concentration was simply due to the availability of more amine for $CO_2$ transport. Upon $CO_2$-amine reaction, various ionic species are formed, such as carbamates, protonated amines and zwitterions. Another possible reason is that these ion species have occupied sites or void spaces that should be used for $CO_2$ transport. As a result, the transport due to solution-diffusion decreases [72,73]. Adding DEA also facilitates $CO_2$ transportation through membranes. Facilitated $CO_2$ transport is a globally non-reactive system, i.e. a system that acts strictly as a passive transport medium and not as a chemical reactor under stable conditions [74]. At a steady state, $CO_2$ transport is mediated by the presence of DEA shuttling back and forth across membrane thickness. DEA on the membrane feed side responds with $CO_2$ to species of carbamates and protonated amines. These species then spread to the membrane's permeate side, where the desorption or reverse reaction is high, releasing $CO_2$. Also, DEA is produced and cycles back to the feed side, where the process is repeated [75].

The effect of DEA concentration on permeance has been well described in the cited literature [75]. It was found that at 20–30 wt% DEA concentration, the permeance of $CO_2$ did not greatly increase. This is due to the trade-off between the favourable facilitation effect of high DEA concentration and the reduction of both ionic species diffusivity and $CO_2$ solubility at high DEA concentrations. Also, the permeance is decreased with the increase of amine concentration because the ionic strength of the membranes increases at higher DEA concentration. Therefore, in this study, DEA concentration was limited to 15 wt%.

## 3.6. Comparison of results with literature

The $CO_2$-$CH_4$ separation factor of investigated composite amine mixed matrix was compared with the results reported in the literature, table 4. It is worth mentioning that no study is reported in the literature on composite amine MMM for $CO_2$-$CH_4$ separation. Hence, the comparison is made with composite MMMs. It shows that the addition of DEA as a carrier enhanced the performance of membrane and non-woven fabric as a support to improve the stability of membranes at high pressure. It was found that the performance of amine composite MMM is comparable with the literature.

# 4. Performance evaluation of amine composite mixed matrix membranes by Maxwell model

The Maxwell model is selected because of its mathematical simplicity and capability to integrate and account for all the physical components of MMMs (such as matrix and filler permeability, filler's volume fraction, where the last parameter plays a critical role in membrane performance). Moreover, it is easy to compute and uses a lesser number of assumptions than other similar models for MMMs [79–81]. However, the Maxwell model is unable to incorporate the effect of alkanolamine in its conventional form. Therefore, a slight modification was performed that allows the Maxwell model to

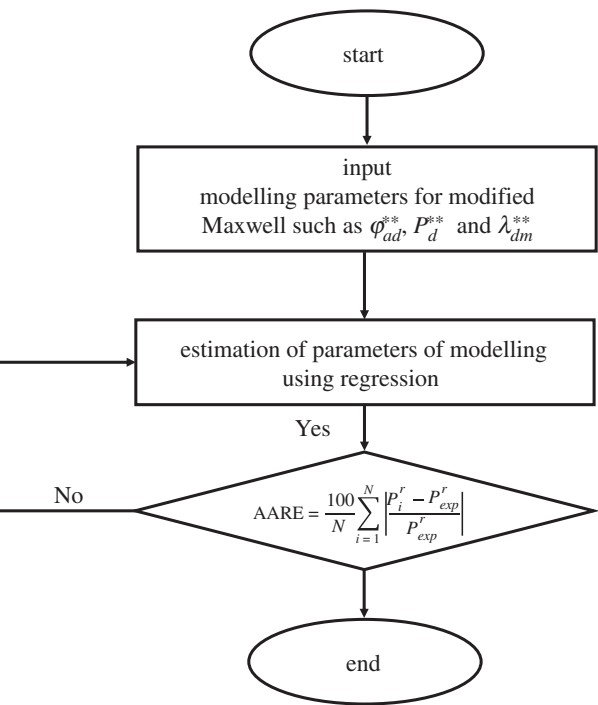

**Figure 10.** Flow chart for the modelling of amine composite MMMs at high pressure.

include the effect of alkanolamine concentration in the performance of the alkanolamine based MMMs. In the Maxwell model, the volume fraction of filler ($\varphi_d$) is a parameter that can play a vital role in predicting the performance of MMM. But in this study, the volume fraction of amine also played a key role in performance enhancement. Therefore, the volume fraction term needs to be modified to evaluate the performance of amine composite MMMs.

The experimental and literature parameters used for the model development are volume fraction of amine $\varphi_a$, the volume fraction of dispersed phase (CMS) $\varphi_d$, the combined volume fraction of filler and DEA $\varphi_{ad}^{**}$, facilitation parameter $F_c$, estimated permeation of filler against pressure $P_d^{**}$, the permeability of polymer matrix $P_m$, and ratio of predicted filler permeability $P_d^{**}$ to the permeability of polymer matrix $P_m$, $\lambda_{dm}^{**}$. The parameters listed above are related to each other as per the equations given below.

$$\varphi_{ad}^{**} = (\varphi_a + \varphi_d) + F_c. \tag{4.1}$$

The volume fraction of CMS and amine is dependent on the experimental studies and has been determined from open literature for amine mixed matrix membranes. As for $F_c$, the value was regressed to experimental values in this study by using the Delaunay Triangulation from the literature [27]. $F_c$ is required in this study to account for the shape factor of the amine in the membrane structure. Thus, $\varphi_{ad}^{**}$ is a combination of three parameters, as stated in equation (4.3).

Usually, the value $\lambda_{dm}^{**}$ is determined at a constant pressure. However, the operating pressure in this study varies. Therefore, the value is regressed to obtain the values at operating pressures used in this study. The fitting and optimization procedures were conducted on the Maxwell model against the high-pressure experimental data to produce the results for $\lambda_{dm}^{**}$. The $P_m$ value, on the other hand, is available through experimental work and varies with pressure. Once the $\lambda_{dm}^{**}$ values are determined, $P_d^{**}$ the predicted permeability of filler against pressure is obtained using equation (4.4).

$$\lambda_{dm}^{**} = \frac{P_d^{**}}{P_m}. \tag{4.2}$$

The addition of the new parameter $\varphi_{ad}^{**}$ in the basic Maxwell equation, based on the electrical or thermal conductivity model, leads to the modified Maxwell equation. Applying all the parameters in equation (4.3) results in determining the relative permeability through modelling. Figure 10 provides a

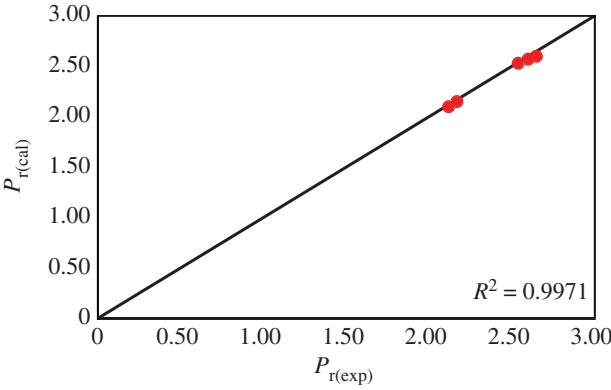

**Figure 11.** $CO_2$ relative permeability (experimental) versus relative permeability (calculated) by the modified Maxwell model.

**Table 5.** Estimated model parameters for fitting the high-pressure experimental data of $CO_2$ separation through the amine composite mixed matrix membranes.

| pressure (bar) | $\varphi_a$ | $\varphi_d$ | $CO_2$ permeance (GPU) CM | CM-C10D15 | Fc [27] | $\varphi_{ad}^{**}$ | $\lambda_{dm}^{**}$ | $P_d^{**}$ | relative permeability $P_{r(exp)}$ | $P_{r(cal)}$ | AARE % |
|---|---|---|---|---|---|---|---|---|---|---|---|
| 10 | 0.05 | 0.13 | 180.5 | 469.31 | 0.27 | 0.45 | 10.66 | 1924.45 | 2.60 | 2.57 | 1.15 |
| 15 |  |  | 161.83 | 411.67 |  |  | 10.07 | 1630.05 | 2.54 | 2.53 | 0.39 |
| 20 |  |  | 143.66 | 380.52 |  |  | 11.24 | 1615.08 | 2.65 | 2.60 | 1.89 |
| 25 |  |  | 137.35 | 297.97 |  |  | 5.83 | 800.50 | 2.17 | 2.15 | 0.92 |
| 30 |  |  | 142.21 | 300.84 |  |  | 5.44 | 773.90 | 2.12 | 2.10 | 0.94 |

graphical representation of the algorithm used for the modelling.

$$P_r = \frac{2(1 - \varphi_{ad}^{**}) + (1 + 2\varphi_{ad}^{**})\lambda_{dm}^{**}}{(2 + \varphi_{ad}^{**}) + (1 - \varphi_{ad}^{**})\lambda_{dm}^{**}}. \tag{4.3}$$

The percent absolute average relative error (AARE%) between the $P_{r(cal)}$ and experimental relative permeability $P_{r(exp)}$ values were determined by equation (4.6).

$$\text{AARE\%} = \frac{100}{N}\sum_{i=1}^{N}\left|\frac{P_i^r - P_{exp}^r}{P_{exp}^r}\right|. \tag{4.4}$$

The model parameters of equation (4.3) are tabulated in table 5. The relative permeability was calculated by putting estimated model parameters for 10 wt% CMS loading and 15 wt% addition of DEA in polymer matrix for the pressure range of 10–30 bar in equation (4.3). It was noted that the calculated relative permeability is in good agreement with experimental relative permeabilities. The AARE% is in the range of 0.39 to 1.89%. Moreover, the experimental and calculated relative permeability was compared to graphically and presented in figure 11. It has shown a good agreement with $R^2 = 0.99$.

A very small difference was found in the values of calculated and experimental values of relative permeabilities. This difference is because the basic Maxwell model was developed for the prediction of electrical conductivity and later further modified for the mixed matrix membranes. The Maxwell model is developed for two (polymer and filler) phase system and low-pressure applications. In this study, the developed amine composite mixed matrix membranes have three (polymer + filler + amine) phases for high-pressure gas transport. Another reason could be the gas transport mechanism [23,82,83]. The developed composite amine mixed matrix membranes follow the solution diffusion through the polymer matrix, molecular sieving through CMS, and facilitated transport through DEA [84].

On the other hand, the Maxwell model obeys the solution diffusion and molecular sieving mechanisms. Beyond these reasons, the modified Maxwell model gives a good prediction of gas transport with small AARE%. In figure 11, the $R^2$ value shows that the modified model can predict the performance of composite amine mixed matrix membranes satisfactory for carbon dioxide.

So, based on results, it is concluded that the modified Maxwell model is working satisfactorily at high pressure 10 wt% of CMS with constant concertation of DEA. But more studies are needed to understand the detailed geometry of composite amine mixed matrix membranes.

# 5. Conclusion

Asymmetric amine composite mixed matrix membranes with CMS, DEA and PES were synthesized and tested for $CO_2$-$CH_4$ separation. The addition of DEA has enhanced $CO_2$-$CH_4$ separation performance, and a more than twofold increment has been observed in $CO_2$ permeance. Similarly, approximately threefold increment has been found in $CO_2$-$CH_4$ selectivity as compared to the pure PES membrane. The FESEM analysis confirmed the smooth and defect-free surface and typical finger-like and sponge-like pore structures, which help to reduce the transport resistance and to provide enough mechanical strength, respectively. The robust mechanical properties of the amine composite MMMs may be attributed to the strong interactions between the PES matrix, CMS, and DEA, in addition to homogeneous dispersion and adhesion of CMS. The modified Maxwell model, after including the optimized $\lambda_{dm}^{**}$ along with facilitation factor $F_c$ and predicted permeability of dispersed phase $P_d^{**}$, has been developed to predict the performance of composite amine MMMs at high pressure. The results exhibited that the model predicted the relative permeability with $R^2 = 0.99$.

Data accessibility. All data associated with the study, including raw gas separation data, have been deposited in the Dryad Digital Repository https://doi.org/10.5061/dryad.dfn2z34xv. Additionally, the FTIR spectra for table 2 has been given as electronic supplementary material with the manuscript.

Authors' contributions. N.A.Bt.F. and N.A. carried out the experiment. R.N. wrote the manuscript and analysed the results with the support of other authors. D.F.Bt.M. and H.A.M. helped supervise the project and analysis and interpretation of data. H.M. conceived the original idea and supervised the project. Z.M. also supervised the project with H.M.

Competing interests. The authors declare that they have no conflict of interest.

Funding. Yayasan UTP (YUTP) fundamental research grant, cost centre number 0153AA-H02.

Acknowledgements. The authors acknowledge the technical and financial support provided by Universiti Teknologi PETRONAS, Malaysia, and Yayasan UTP (YUTP) fundamental research grant cost centre number 0153AA-H02, respectively.

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
