## [Reviewer comments · Royal Society Open Science]

Review History

RSOS-200795.R0 (Original submission)

Review form: Reviewer 1

Is the manuscript scientifically sound in its present form?

Yes

Are the interpretations and conclusions justified by the results?

Yes

Is the language acceptable?

No

Do you have any ethical concerns with this paper?

No

Have you any concerns about statistical analyses in this paper?

No

Recommendation?

Accept with minor revision (please list in comments)

Comments to the Author(s)

The manuscript entitled "High-Pressure CO₂-CH₄ Separation through Composite Amine Mixed Matrix Membranes: Synthesis, Characterization and Performance Prediction" studies the high pressure gas separation performance of amine based mixed matrix membranes. Overall the structure of manuscript is in line and major points of concerns are already discussed. Manuscript has sufficient contribution to be accepted in this journal followed by minor revisions. Following points will help authors in improving their manuscript.

1. Language of the manuacript needs an extensive revision. There are some words missing, grammatical mistakes and vocabulary mistakes.
2. Abstract should highlight the particular novelty of this work.
3. Introduction part is too long, please focus on the problem and proposed solution
4. How authors will justify the use of Maxwell model for this type of membrane. Which parameter of maxwell model they have considered important and key player while predicting the data.
5. Explanation of membrane morphology is very general. Specific reasons and justifications are required for membrane micro structures
6. How experimental parameters were selected? Why only 30 bar? Why not 50 bar?
7. Modelling part is quite confusing and was not explained well.
8. Results must be discussed comprehensively with justifications and references
9. Gas separation results must be compared with literature to highlight the importance of these membranes
10. More references are needed for the conclusions drawn by authors based on characterization results
11. How polymer concentration and amines concentration was selected in membrane development and how change in concentration will affect the performance of membranes?

Review form: Reviewer 2

Is the manuscript scientifically sound in its present form?

Yes

Are the interpretations and conclusions justified by the results?

Yes

Is the language acceptable?

Yes

Do you have any ethical concerns with this paper?

No

Have you any concerns about statistical analyses in this paper?

No

Recommendation?

Accept with minor revision (please list in comments)

Comments to the Author(s)

The article is well prepared, properly organised and scientifically sound with comprehensive discussion on the adopted methodology and obtained results. However, there are certain concerns that need to be addressed to enhance the quality of the article.

General comments,

- 1) In my point of view the title of the article should be reconsidered. The words "performance prediction" do not seem appropriate here, as the authors actually evaluated the membranes performance rather than predicting this via models.
- 2) The authors need to standardize the abbreviations, symbols and presenting various units. For example, the weight percent is mentioned with "weight %, wt. % and wt %", the authors should adopt one style and follow it through out the manuscript. Similarly, Tg is mentioned in normal letters as well as in Italic styles. Furthermore, some times space is used between a value and units (30 bar), sometimes its mentioned without space (30bar).
- 3) The authors should consider English proofreading.
- 4) The captions of the Figure 6 need modification with sub-numbering.
- 5) The authors used "CO₂/CH₄ and CO₂-CH₄" terms throughout the manuscript, this also require standardization in the manuscript.
- 6) Caption of table 3 is not related to the entries in the table.
- 7) Table 2, FTIR spectra is presented. The authors need to provide the references in this table.

Technical Comments;

- 1) It is mentioned that pure PES membrane possessed sponge like morphology that provide mechanical strength (Page 8). However, mechanical properties are far low compared to CM-C10D10 membrane (Table 3). Authors need to further elaborate this phenomenon. Most probable reason in my point of view is, the morphology of the pure membrane is not merely sponge like only but combination of both sponge and finger like as shown in Figure 4.
- 2) Pg 15, L 11, The authors speculate that the addition of DEA results in tensile strength decrement. But according to Table 3, the tensile strength values of DEA containing membranes are higher than pure PES membrane. The authors should support their speculations with already published works.
- 3) As in Figure 8, CO₂ permanence and CO₂/CH₄ separation both keep on increasing with DEA addition. Why authors do not consider to go beyond 15 wt. % DEA in order to check the maximum potential of the DEA in PES polymer.
- 4) Pg 14, "Upon CO₂-amine reaction, various ionic species are formed, such as carbamates, protonated amines, and zwitterions". If this is the probable reason for CO₂ enhanced permeance properties in DEA incorporated membranes, then authors should further elaborate this theory from the previous works and relate this with their findings.

Decision letter (RSOS-200795.R0)

Dear Professor Mukhtar:

Title: High-Pressure CO₂-CH₄ Separation through Composite Amine Mixed Matrix Membranes: Synthesis, Characterization and Performance Prediction
 Manuscript ID: RSOS-200795

The editor assigned to your manuscript has now received comments from reviewers. We would like you to revise your paper in accordance with the referee and Subject Editor suggestions which

can be found below (not including confidential reports to the Editor). Please note this decision does not guarantee eventual acceptance.

Please submit your revised paper before 25-Jul-2020. Please note that the revision deadline will expire at 00.00am on this date. If we do not hear from you within this time then it will be assumed that the paper has been withdrawn. In exceptional circumstances, extensions may be possible if agreed with the Editorial Office in advance. We do not allow multiple rounds of revision so we urge you to make every effort to fully address all of the comments at this stage. If deemed necessary by the Editors, your manuscript will be sent back to one or more of the original reviewers for assessment. If the original reviewers are not available we may invite new reviewers.

On behalf of the Subject Editor Professor Anthony Stace and the Associate Editor Dr Chaohua Cui.

RSC Associate Editor:

Comments to the Author:

The manuscript requires revision with respect to the language used. I therefore suggest that you ask a native English speaker or equivalent to assist you with correcting the spelling, grammar, word use, and punctuation throughout your manuscript.

RSC Subject Editor:

Comments to the Author:

(There are no comments.)

Reviewers' Comments to Author:

Reviewer: 1

Comments to the Author(s)

The manuscript entitled "High-Pressure CO₂-CH₄ Separation through Composite Amine Mixed Matrix Membranes: Synthesis, Characterization and Performance Prediction" studies the high pressure gas separation performance of amine based mixed matrix membranes. Overall the structure of manuscript is in line and major points of concerns are already discussed. Manuscript has sufficient contribution to be accepted in this journal followed by minor revisions. Following points will help authors in improving their manuscript.

1. Language of the manuacript needs an extensive revision. There are some words missing, grammatical mistakes and vocabulary mistakes.
2. Abstract should highlight the particular novelty of this work.
3. Introduction part is too long, please focus on the problem and proposed solution
4. How authors will justify the use of Maxwell model for this type of membrane. Which parameter of maxwell model they have considered important and key player while predicting the data.
5. Explanation of membrane morphology is very general. Specific reasons and justifications are required for membrane micro structures
6. How experimental parameters were selected? Why only 30 bar? Why not 50 bar?
7. Modelling part is quite confusing and was not explained well.
8. Results must be discussed comprehensively with justifications and references
9. Gas separation results must be compared with literature to highlight the importance of these membranes
10. More references are needed for the conclusions drawn by authors based on characterization results
11. How polymer concentration and amines concentration was selected in membrane development and how change in concentration will affect the performance of membranes?

Reviewer: 2

Comments to the Author(s)

The article is well prepared, properly organised and scientifically sound with comprehensive discussion on the adopted methodology and obtained results. However, there are certain concerns that need to be addressed to enhance the quality of the article.

General comments,

- 1) In my point of view the title of the article should be reconsidered. The words "performance prediction" do not seem appropriate here, as the authors actually evaluated the membranes performance rather than predicting this via models.
- 2) The authors need to standardize the abbreviations, symbols and presenting various units. For example, the weight percent is mentioned with "weight %, wt. % and wt %", the authors should adopt one style and follow it through out the manuscript. Similarly, T_g is mentioned in normal letters as well as in Italic styles. Furthermore, some times space is used between a value and units (30 bar), sometimes its mentioned without space (30bar).
- 3) The authors should consider English proofreading.
- 4) The captions of the Figure 6 need modification with sub-numbering.
- 5) The authors used "CO₂/CH₄ and CO₂-CH₄" terms throughout the manuscript, this also require standardization in the manuscript.
- 6) Caption of table 3 is not related to the entries in the table.
- 7) Table 2, FTIR spectra is presented. The authors need to provide the references in this table.

Technical Comments;

- 1) It is mentioned that pure PES membrane possessed sponge like morphology that provide mechanical strength (Page 8). However, mechanical properties are far low compared to CM-C10D10 membrane (Table 3). Authors need to further elaborate this phenomenon. Most probable

reason in my point of view is, the morphology of the pure membrane is not merely sponge like only but combination of both sponge and finger like as shown in Figure 4.

2) Pg 15, L 11, The authors speculate that the addition of DEA results in tensile strength decrement. But according to Table 3, the tensile strength values of DEA containing membranes are higher than pure PES membrane. The authors should support their speculations with already published works.

3) As in Figure 8, CO₂ permanence and CO₂/CH₄ separation both keep on increasing with DEA addition. Why authors do not consider to go beyond 15 wt. % DEA in order to check the maximum potential of the DEA in PES polymer.

4) Pg 14, "Upon CO₂-amine reaction, various ionic species are formed, such as carbamates, protonated amines, and zwitterions". If this is the probable reason for CO₂ enhanced permeance properties in DEA incorporated membranes, then authors should further elaborate this theory from the previous works and relate this with their findings.

Author's Response to Decision Letter for (RSOS-200795.R0)

See Appendix A.

RSOS-200795.R1 (Revision)

Review form: Reviewer 1

Is the manuscript scientifically sound in its present form?

Yes

Are the interpretations and conclusions justified by the results?

Yes

Is the language acceptable?

Yes

Do you have any ethical concerns with this paper?

No

Have you any concerns about statistical analyses in this paper?

No

Recommendation?

Accept as is

Comments to the Author(s)

The manuscript is revised well and it can be accepted in its present form.

Review form: Reviewer 2

Is the manuscript scientifically sound in its present form?

Yes

Are the interpretations and conclusions justified by the results?

Yes

Is the language acceptable?

Yes

Do you have any ethical concerns with this paper?

No

Have you any concerns about statistical analyses in this paper?

No

Recommendation?

Accept as is

Comments to the Author(s)

The authors have addressed all the comments satisfactorily after the 1st revision. In best of my understanding, the manuscript should now be accepted in its present form.

Decision letter (RSOS-200795.R1)

Dear Professor Mukhtar:

Title: Composite Amine Mixed Matrix Membranes for High-Pressure CO₂-CH₄ Separation: Synthesis, Characterization and Performance Evaluation

Manuscript ID: RSOS-200795.R1

It is a pleasure to accept your manuscript in its current form for publication in Royal Society Open Science. The chemistry content of Royal Society Open Science is published in collaboration with the Royal Society of Chemistry.

Yours sincerely,

Dr Laura Smith

Publishing Editor, Journals

Royal Society of Chemistry

Thomas Graham House

Science Park, Milton Road

Cambridge, CB4 0WF

Royal Society Open Science - Chemistry Editorial Office

On behalf of the Subject Editor Professor Anthony Stace and the Associate Editor Dr Chaohua Cui.

RSC Associate Editor:
Comments to the Author:
(There are no comments.)

RSC Subject Editor:
Comments to the Author:
(There are no comments.)

Reviewer(s)' Comments to Author:
Reviewer: 2

Comments to the Author(s)
The authors have addressed all the comments satisfactorily after the 1st revision. In best of my understanding, the manuscript should now be accepted in its present form.

Reviewer: 1

Comments to the Author(s)
The manuscript is revised well and it can be accepted in its present form.

Appendix A

RESPONSE TO DECISION LETTER ON MANUSCRIPT ID: RSOS-200795

Dear Editors and Reviewers,

Greetings,

We would like to present our profound gratitude for your valuable comments and suggestions. We are of the view that the comments raised, and the suggestions given are very much relevant and necessary to uplift the paper quality. We have addressed each comment/suggestion and revised the manuscript in view of these comments. We hope that the revised manuscript would prove to be in synchrony with the format and standard of this esteemed journal. The detailed responses to the specific comments/suggestions/queries are as follows.

Sr. No	RSC Associate Editor	
	Comment	Response
01	The manuscript requires revision with respect to the language used. I therefore suggest that you ask a native English speaker or equivalent to assist you with correcting the spelling, grammar, word use, and punctuation throughout your manuscript.	The manuscript has been proofread and checked/corrected by native English speakers.

Sr. No.	Reviewer 1	
	Comments	Response
01	Language of the manuscript needs an extensive revision. There are some words missing, grammatical mistakes and vocabulary mistakes	The manuscript has been proofread and checked/corrected by a native English speaker.
02	Abstract should highlight the particular novelty of this work.	The membranes were aimed to operate at higher pressure operations. It has been highlighted in the abstract.
03	Introduction part is too long, please focus on the problem and proposed solution.	Thanks for the comment. The introduction has been shortened to address the problem and the proposed solution.
04	How authors will justify the use of Maxwell model for this type of membrane. Which parameter of maxwell model they have considered important and key player while predicting the data.	The justification has been provided in Section 1 and Section 4
05	Explanation of membrane morphology is very general. Specific reasons and justifications are required for membrane micro structures	The specific reasons and justification have been provided in section 3.1 and section 3.1.3
06	How experimental parameters were selected? Why only 30 bar? Why not 50 bar?	Thanks for the comment, the justification has been added in section 3.5.1
07	Modelling part is quite confusing and was not explained well.	The explanation has been improved for better understanding.
08	Results must be discussed comprehensively with justifications and references	New references have been added in results sections for better understanding and clear justifications. Check references [36-39,42,48,49,52,61,63-67,71,72,76-82].
09	Gas separation results must be compared with literature to highlight the importance of these membranes	Comparison has been given in Table 4, section 3.6.
10	More references are needed for the conclusions drawn by authors based on characterization results	New references have been added in characterization sections for better understanding. Check references [36-39,42,48,49,52,61,63-67,71,72,76-82].

11	How polymer concentration and amines concentration was selected in membrane development and how change in concentration will affect the performance of membranes?	Critical polymer concentration was selected for membrane development. Its data has been referred and provided in section 2.2 The amine concentration was based on the weight of the polymer. Its concentration effects have been discussed in Section 3.5.2.
----	---	---

Sr. No.	Reviewer 2	
	General Comments	Response
01	In my point of view the title of the article should be reconsidered. The words "performance prediction" do not seem appropriate here, as the authors actually evaluated the membranes performance rather than predicting this via models.	Agree with the reviewer, and the title has been modified by replacing the "prediction" with "evaluation".
02	The authors need to standardize the abbreviations, symbols and presenting various units. For example, the weight percent is mentioned with "weight %, wt. % and wt %", the authors should adopt one style and follow it throughout the manuscript. Similarly, Tg is mentioned in normal letters as well as in Italic styles. Furthermore, sometimes space is used between a value and units (30 bar), sometimes its mentioned without space (30bar).	Sorry for these mistakes, the whole manuscript has been checked and corrected according to comment.
03	The authors should consider English proofreading.	The manuscript has been proofread and checked/corrected by native English speakers.
04	The captions of the Figure 6 need modification with sub-numbering	Sorry the mistake authors modify the caption of Figure 6
05	The authors used "CO ₂ /CH ₄ and CO ₂ -CH ₄ " terms throughout the manuscript, this also require standardization in the manuscript.	Sorry for these mistakes, the whole manuscript has been checked and corrected according to comment.
06	Caption of table 3 is not related to the entries in the table.	Thanks for the comments, the caption has been changed. It was a typo error.
07	Table 2, FTIR spectra is presented. The authors need to provide the references in this table.	The reference has been provided in the table caption. The spectrum was analyzed with the help of renown book titled " Introduction to Spectroscopy " by Donald L.

		Pavia, Gary M. Lampman, George S. Kriz, James A. Vyvyan
	Technical Comments	
01	It is mentioned that pure PES membrane possessed sponge like morphology that provide mechanical strength (Page 8). However, mechanical properties are far low compared to CM-C10D10 membrane (Table 3). Authors need to further elaborate this phenomenon. Most probable reason in my point of view is, the morphology of the pure membrane is not merely sponge like only but combination of both sponge and finger like as shown in Figure 4.	Agree with the reviewer pure PES membranes have both finger-like and sponge structure. It has been mentioned in section 3.1.1.
02	Pg 15, L 11, The authors speculate that the addition of DEA results in tensile strength decrement. But according to Table 3, the tensile strength values of DEA containing membranes are higher than pure PES membrane. The authors should support their speculations with already published works.	Thanks for the comments, we have modified the discussion. The author wants to say that by the addition of DEA, the tensile strength has been improved. See section 3.4
03	As in Figure 8, CO ₂ permanence and CO ₂ /CH ₄ separation both keep on increasing with DEA addition. Why authors do not consider to go beyond 15 wt. % DEA in order to check the maximum potential of the DEA in PES polymer.	Thanks for the comments the detailed discussion has been provided in section 3.5.2
04	Pg 14, "Upon CO ₂ -amine reaction, various ionic species are formed, such as carbamates, protonated amines, and zwitterions". If this is the probable reason for CO ₂ enhanced permeance properties in DEA incorporated membranes, then authors should further elaborate this theory from the previous works and relate this with their findings.